# Coupling of Higgs and Leggett modes in non-equilibrium superconductors

H. Krull[1], N. Bittner[2], G.S. Uhrig[1], D. Manske[2] & A.P. Schnyder[2]

In equilibrium systems amplitude and phase collective modes are decoupled, as they are mutually orthogonal excitations. The direct detection of these Higgs and Leggett collective modes by linear-response measurements is not possible, because they do not couple directly to the electromagnetic field. In this work, using numerical exact simulations we show for the case of two-gap superconductors, that optical pump–probe experiments excite both Higgs and Leggett modes out of equilibrium. We find that this non-adiabatic excitation process introduces a strong interaction between the collective modes, which is absent in equilibrium. Moreover, we propose a type of pump–probe experiment, which allows to probe and coherently control the Higgs and Leggett modes, and thus the order parameter directly. These findings go beyond two-band superconductors and apply to general collective modes in quantum materials.

[1] Lehrstuhl für Theoretische Physik I, Technische Univerität Dortmund, Otto-Hahn Strasse 4, D-44221 Dortmund, Germany. [2] Max-Planck-Institut für Festkörperforschung, Heisenbergstrasse 1, D-70569 Stuttgart, Germany. Correspondence and requests for materials should be addressed to A.S. (email: a.schnyder@fkf.mpg.de).

ollective excitation modes are a characteristic feature of symmetry-broken phases of matter. These collective excitations are due to amplitude and phase fluctuations of the order parameters, which are decoupled in equilibrium systems, as they represent mutually orthogonal excitations. The properties of collective modes are of fundamental interest, as they are a distinguishing feature of any symmetry-broken phase, such as superconductors, charge-density waves or antiferromagnets. For example, superconductors exhibit an amplitude Higgs mode and a phase mode, which are the radial and angular excitations in the Mexican-hat potential of the free energy. In two-band superconductors there exists in addition a Leggett phase mode[1], which corresponds to collective fluctuations of the interband phase difference.

As we show in this work, out-of-equilibrium excitations of symmetry-broken phases lead to a direct coupling between phase and amplitude modes, an effect which is absent in equilibrium systems. Furthermore, we demonstrate that ultrafast pump–probe measurements allow to directly probe and coherently control collective excitation modes. Pump–probe measurements have recently become a key tool to probe the temporal dynamics and relaxation of quantum materials[2–13]. In particular, this technique has been used to measure the oscillations of the amplitude Higgs mode of the one-gap superconductor NbN. It has been shown, both theoretically[14–28] and experimentally[2–4], that a short intense laser pulse of length $\tau$ much shorter than the dynamical time scale of the superconductor $\tau_\Delta \simeq h/(2|\Delta|)$ induces oscillations in the order parameter amplitude at the frequency $\omega_H = 2\Delta_\infty/\hbar$, with $\Delta_\infty$ the asymptotic gap value.

Although non-equilibrium collective modes in conventional single-gap superconductors are well understood, the investigation of collective excitations in unconventional non-equilibrium superconductors with multiple gaps, such as $MgB_2$ or iron pnictides, is still in its infancy[29–31]. These multicomponent superconductors have a particularly rich spectrum of collective excitations[1,32,33]. In this study, we simulate the pump–probe process in a two-gap superconductor using a semi-numerical approach based on the density-matrix formalism. This method is exact for mean-field Hamiltonians[15,21], captures the coupling between the superconductor and the electromagnetic field of the pump laser at a microscopic level and allows for the calculation of the pump–probe conductivity, as measured in recent experiments[2,3]. Two-gap superconductors exhibit besides the amplitude Higgs[34] and the phase modes[35,36], also a Leggett mode[1], which results from fluctuations of the relative phase of the two coupled gaps, that is, equal but opposite phase shifts of the two-order parameters, see Fig. 1b. In equilibrium superconductors, the Higgs and Leggett modes are decoupled, as they correspond to mutually orthogonal fluctuations. In contrast to the phase mode, both Higgs and Leggett modes are charge neutral and therefore do not couple to the electromagnetic field[37]. As a consequence, these excitations cannot be detected directly with standard linear-response-type measurements. Observation of these modes has only been possible in special circumstance, for example, when they couple to another order parameter, such as in charge density wave systems[38–41].

Here we show that in a pump–probe experiment both Leggett and Higgs modes can be excited out of equilibrium, and directly observed as oscillations in the absorption spectra at their respective frequencies. We find that the non-adiabatic excitation process of the pump pulse induces an intricate coupling between the two charge-neutral modes, which pushes the frequency of the Leggett mode below the continuum of two-particle excitations. Moreover, the frequencies of the Leggett and Higgs modes and the coupling between them can be controlled by the fluence of the pump pulse. Hence, by adjusting the laser intensity the two

modes can be brought into resonance, which greatly enhances their oscillatory signal in the pump–probe absorption spectra.

## Results

**Pump-excitation process**. In pump–probe measurements, the pump laser pulse excites a high density of quasiparticles above the gap of the order parameter, thereby modifying the Mexican-hat potential of the free energy $\mathcal{F}$ (Fig. 1). As a result, the amplitude of the order parameter decreases, reducing the minimum of the free energy. If the pump-pulse-induced changes in $\mathcal{F}$ occur on a faster time scale than the intrinsic response time of the symmetry-broken state, the collective modes start to oscillate at their characteristic frequencies about the new free-energy minimum (see Fig. 1a). In this work we study this non-adiabatic excitation mechanism for two-band superconductors perturbed by a short and intense pump pulse.

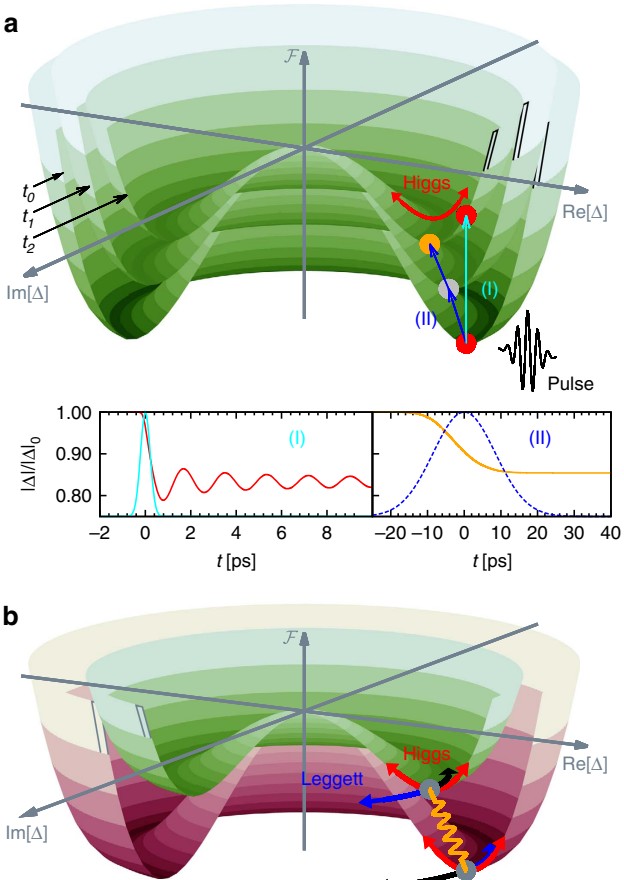

**Figure 1 | Illustration of Leggett and Higgs modes.** (**a**) Illustration of the excitation process for a one band superconductor. The pump laser pulse modifies the free energy $\mathcal{F}$ on different time scales depending on the pulse duration $\tau$. For $\tau \gg h/(2|\Delta|)$ the superconductor can follow the change in $\mathcal{F}$ adiabatically, resulting in a monotonic lowering of the order parameter $|\Delta|$ (orange trace in inset (II)). For short pulses with $\tau \lesssim h/(2|\Delta|)$, on the other hand, the superconductor is excited in a non-adiabatic manner, which results in oscillations of $|\Delta|$ about the new minimum of $\mathcal{F}$ (red trace in inset (I)). The blue and cyan lines in the two insets represent the Gaussian profiles of the pump pulses. (**b**) Effective free-energy landscape $\mathcal{F}$ for a two-gap superconductor, with green and red representing the Mexican-hat potentials of the smaller and larger gaps, respectively. The amplitude Higgs modes and the phase modes are indicated by red and blue/black arrows, respectively. The Leggett mode corresponds to out-of-phase fluctuations of the phase difference between the two gaps.

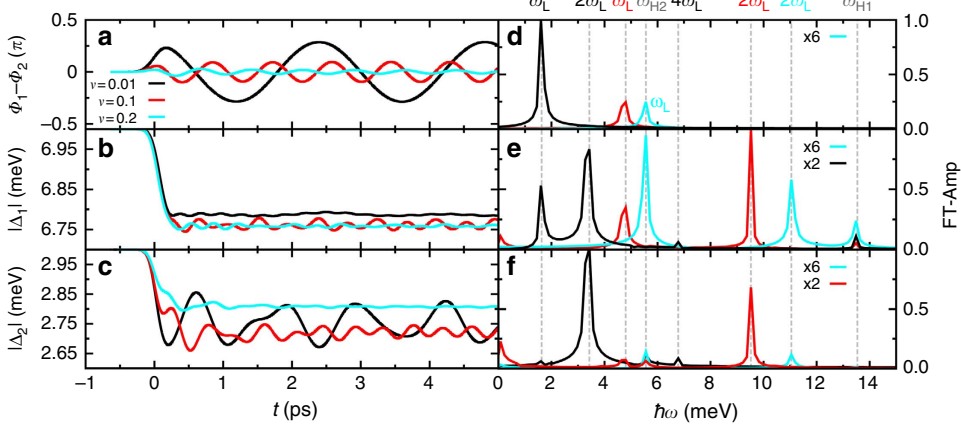

**Figure 2 | Leggett phase mode and amplitude Higgs mode oscillations.** Numerical simulation of the gap dynamics of a two-gap superconductor after a non-adiabatic excitation by a short intense laser pulse of width $\tau = 0.4\,\text{ps}$, pump energy $\hbar\omega_0 = 8\,\text{meV}$ and light-field amplitude $|\mathbf{A}_0| = 10 \times 10^{-8}\,\text{Js}\,(\text{Cm})^{-1}$. (**a**) Phase difference $\Phi_1-\Phi_2$ between the two gaps as a function of time $t$ for various interband coupling strengths $\upsilon$. (**d**) Fourier spectrum of the oscillations in **a**. The frequency of the non-equilibrium Leggett mode oscillation is indicated by $\omega_L$. (**b,c**) Gap amplitudes $|\Delta_1|$ and $|\Delta_2|$ as a function of time $t$ for different interband couplings $\upsilon$. (**e,f**) Fourier spectra of the amplitude mode oscillations in **b,c**, which display the following frequencies: $\omega_{H1}$ and $\omega_{H2}$ are the frequencies of the non-equilibrium Higgs modes of gap $\Delta_1$ and $\Delta_2$, respectively; $\omega_L$ is the frequency of the non-equilibrium Leggett mode; and higher harmonics of the non-equilibrium Leggett mode are denoted by $2\omega_L$ and $4\omega_L$.

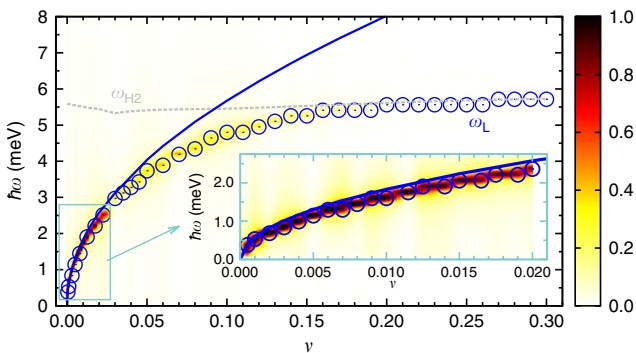

**Figure 3 | Leggett phase mode oscillations versus relative interband coupling.** Fourier spectrum of the phase mode $\Phi_1-\Phi_2$ as a function of relative interband coupling $\upsilon$ for a two-band superconductor perturbed by the same laser pulse as in Fig. 2. The amplitude of the phase fluctuations is indicated by the colour scale with dark red and light yellow representing the highest and lowest amplitudes, respectively. The blue open circles mark the frequency of the non-equilibrium Leggett mode $\omega_L$. The blue solid line represents the frequency of the equilibrium Leggett mode described by equation (2). The dashed grey line indicates the frequency of the Higgs mode $\omega_{H2}$, which coincides with the boundary to the continuum of Bogoliubov quasiparticle excitations, given by twice the asymptotic gap value of the second band $2\Delta_2^\infty$. The inset shows a zoom-in of the blue frame in the main panel.

The Hamiltonian describing the two-band superconductor coupled to the pump laser field is given by $H = H_{\text{BCS}} + H_{\text{laser}}$, with the two-band Bardeen-Cooper-Schrieffer (BCS) mean-field Hamiltonian

$$H_{\text{BCS}} = H_0 + \sum_{\mathbf{k}\in\mathcal{W}}\sum_{l=1}^{2}(\Delta_l c_{\mathbf{k}l\uparrow}^\dagger c_{-\mathbf{k}l\downarrow}^\dagger + \Delta_l^* c_{-\mathbf{k}l\downarrow}c_{\mathbf{k}l\uparrow}), \quad (1)$$

where $H_0 = \sum_{\mathbf{k}l\sigma}\varepsilon_{\mathbf{k}l}c_{\mathbf{k}l\sigma}^\dagger c_{\mathbf{k}l\sigma}$ denotes the normal state Hamiltonian and $c_{\mathbf{k}l\sigma}^\dagger$ creates electrons with momentum $\mathbf{k}$, band index $l$ and spin $\sigma$. The first sum in equation (1) is taken over the set $\mathcal{W}$ of momentum vectors with $|\varepsilon_{\mathbf{k}l}| \leq \hbar\omega_c = 50\,\text{meV}$, $\omega_c$ being the cutoff frequency. The gaps $\Delta_1$ and $\Delta_2$ in the two bands are

determined at each temporal integration step from the BCS gap equations with the attractive intraband pairing interactions $V_1$ and $V_2$, and the interband coupling $V_{12} = \upsilon V_1$. Motivated by the numbers for $MgB_2$ (ref. 42), we fix $V_1$ and $V_2$ such that the gaps in the initial state take on the values $\Delta_1(t_i) = 7\,\text{meV}$ and $\Delta_2(t_i) = 3\,\text{meV}$, and study the dynamics of the two-gap superconductor as a function of the relative interband coupling $\upsilon$. $H_{\text{laser}}$ represents the interaction of the pump laser with the superconductor and contains terms linear and quadratic in the vector potential of the laser field, which is of Gaussian shape with central frequency $\hbar\omega_0 = 8\,\text{meV}$, pulse width $\tau = 0.4\,\text{ps}$ and light-field amplitude $|\mathbf{A}_0|$. We determine the dynamics of Hamiltonian (1) by means of the density matrix approach and solve the resulting equations of motion using Runge–Kutta integration (see Methods).

**Pump response.** Pumping the two-band superconductor with a short laser pulse of length $\tau \ll \tau_\Delta$ excites a non-thermal distribution of Bogoliubov quasiparticles above the gaps $\Delta_i$, which in turn leads to a rapid, non-adiabatic change in the free-energy landscape $\mathcal{F}$ (Fig. 1). As a result, the collective modes of the superconductor start to oscillate about the new minima of $\mathcal{F}$. This is clearly visible in Fig. 2, which shows the temporal evolution of the gap amplitudes $|\Delta_i|$ and of the phase difference $\Phi_1-\Phi_2$ between the two gaps. From the Fourier spectra in Figs 2d–f we can see that three different modes (and their higher harmonics) are excited at the frequencies $\omega_{H1}$, $\omega_{H2}$ and $\omega_L$. The two modes at $\omega_{H1}$ and $\omega_{H2}$ only exist in the dynamics of $\Delta_1(t)$ and $\Delta_2(t)$, respectively, and their peaks are located at the energy of the superconducting gaps $\omega_{Hi} = 2|\Delta_i^\infty|/\hbar$, where $\Delta_i^\infty$ denotes the asymptotic gap value[14–20]. This holds for all parameter regimes, even as the laser fluence is increased far beyond the linear absorption region (see Fig. 5). We therefore assign the peaks at $\omega_{H1}$ and $\omega_{H2}$ to the Higgs amplitude modes of the two gaps. The higher Higgs mode $\omega_{H1}$ is strongly damped, because it lies within the continuum of Bogoliubov quasiparticle excitations, which is lower bounded by $2\Delta_2^\infty$. For the lower mode $\omega_{H2}$, on the other hand, the decay channel to quasiparticles is small, as $\omega_{H2}$ is at the continuum threshold. This is similar to the non-equilibrium Higgs mode of the single-gap superconductor NbN, whose oscillations have recently been observed over a time period of about 10 ps by pump–probe measurements[2,3].

Interestingly, two-band superconductors exhibit a third collective mode besides the two Higgs modes at a frequency $\omega_L$ below the quasiparticle continuum. This mode is most clearly visible in the dynamics of the phase difference $\Phi_1-\Phi_2$ (Fig. 2a) and displays a striking dependence on interband coupling strength $\upsilon$. With decreasing $\upsilon$ its frequency rapidly decreases, whereas its intensity grows. In the limit of vanishing $\upsilon$, however, the third mode $\omega_L$ is completely absent. We thus identify $\omega_L$ as

the Leggett phase mode, that is, as equal but opposite oscillatory phase shifts of the two coupled gaps. Remarkably, the Leggett phase mode is also observable in the time dependence of the gap amplitudes $\Delta_1(t)$ and $\Delta_2(t)$ (Figs 2b,c), which indicates that Higgs and Leggett modes are coupled in non-equilibrium superconductors.

To obtain a more detailed picture, we plot in Figs 3 and 4 the energies of the amplitude and phase mode oscillations against the relative interband coupling $\upsilon$. This reveals that for small $\upsilon$ the non-equilibrium Leggett mode $\omega_L$ shows a square root increase, which is in good agreement with the equilibrium Leggett frequency[1,43]

$$\omega_L^0 = 2\sqrt{\Delta_1^\infty \Delta_2^\infty \frac{\upsilon}{V_2 - \upsilon^2 V_1}\left(\frac{1}{\rho_1} + \frac{1}{\rho_2}\right)},\qquad (2)$$

where $\rho_1$ and $\rho_2$ denote the density of states on the two bands. Indeed, as displayed by the inset of Fig. 3, equation (2) represents an excellent parameter-free fit to the numerical data at low $\upsilon$. For larger $\upsilon$, on the other hand, the non-equilibrium Leggett mode deviates from the square root behaviour of equation (2). That is, as $\omega_L$ approaches the Bogoliubov quasiparticle continuum, it is repelled by the lower Higgs mode $\omega_{H2}$, evidencing a strong coupling between them. As a result, the non-equilibrium Leggett mode is pushed below the continuum and remains nearly undamped for a wide range of $\upsilon$, which is considerably broader than in equilibrium. Moreover, owing to the dynamical coupling among the collective modes, $\omega_L$ and its higher harmonics are observable not only in the phase difference $\Phi_1-\Phi_2$ but also in the dynamics of the gap amplitudes $\Delta_i(t)$ (blue and green circles in Fig. 4).

A key advantage of measuring collective modes by pump–probe experiments is that the frequencies of the Higgs modes can be adjusted by the pump fluence. This is demonstrated in Fig. 5, which plots the dynamics of $\Delta_i(t)$ and $\Phi_1-\Phi_2$ as a function of integrated pump pulse intensity $|A_0|^2\tau$. With increasing pump fluence, more Cooper pairs are broken up and superconductivity is more and more suppressed, as reflected in the reduction of the gap amplitudes. At the same time, the frequency of the Higgs oscillations decreases, as it is controlled by the superconducting gaps after pumping. Hence, it is possible to tune the lower Higgs mode $\omega_{H2}$ to resonance with $\omega_L$, which strongly enhances the magnitude of the collective-mode oscillations (Fig. 5a,c,e). A

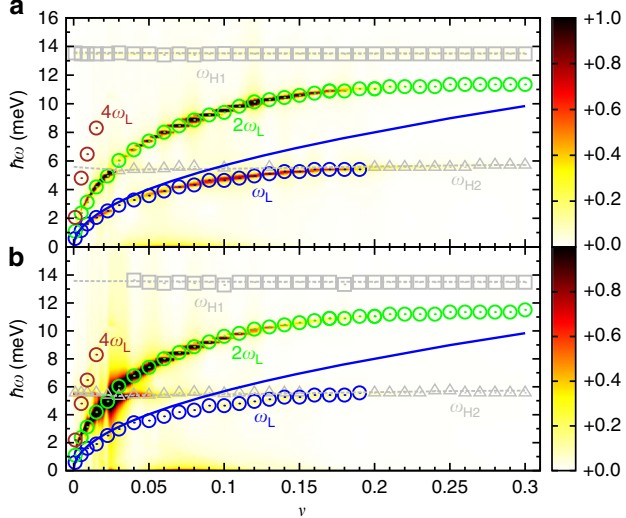

**Figure 4 | Amplitude mode oscillations versus relative interband coupling.** Fourier spectrum of the amplitude mode oscillations as a function of relative interband coupling $\upsilon$ for (**a**) the superconducting gap $\Delta_1$ in the first band and (**b**) the superconducting gap $\Delta_2$ in the second band. The parameters of the laser pump pulse are the same as in Fig. 2. The amplitude of the oscillations is indicated by the colour scale with dark red and light yellow representing the highest and lowest amplitudes, respectively. The open circles represent the frequencies of the non-equilibrium Leggett mode $\omega_L$ and its higher harmonics denoted by $2\omega_L$ and $4\omega_L$. The frequencies of the non-equilibrium Higgs mode of the first and second band, $\omega_{H1}$ and $\omega_{H2}$, are indicated by the grey open squares and triangles, respectively. The blue solid line is the frequency of the equilibrium Leggett mode given by equation (2).

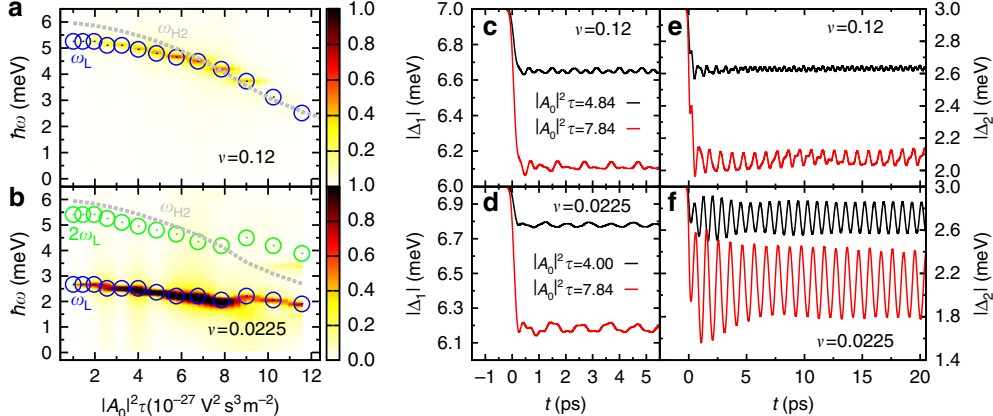

**Figure 5 | Fluence dependence of gap dynamics.** (**a,b**) Fourier spectrum of the phase mode $\Phi_1-\Phi_2$ as a function of laser fluence (integrated pulse intensity) $|A_0|^2\tau$ for two different interband couplings $\upsilon$ with pulse energy $\hbar\omega_0 = 8$ meV and pulse width $\tau = 0.4$ ps. The amplitude of the phase fluctuations is represented by the colour scale with dark red and light yellow indicating high and low amplitudes, respectively. The open circles mark the frequencies of the non-equilibrium Leggett mode $\omega_L$ and its higher harmonic $2\omega_L$. The grey dotted lines display the Higgs mode $\omega_{H2}$, which coincides with the boundary to the continuum of Bogoliubov quasiparticle excitations. (**c–f**) Gap amplitudes $|\Delta_1|$ and $|\Delta_2|$ as a function of time $t$ for two different interband couplings $\upsilon$ and integrated pulse intensities $|A_0|^2\tau$.

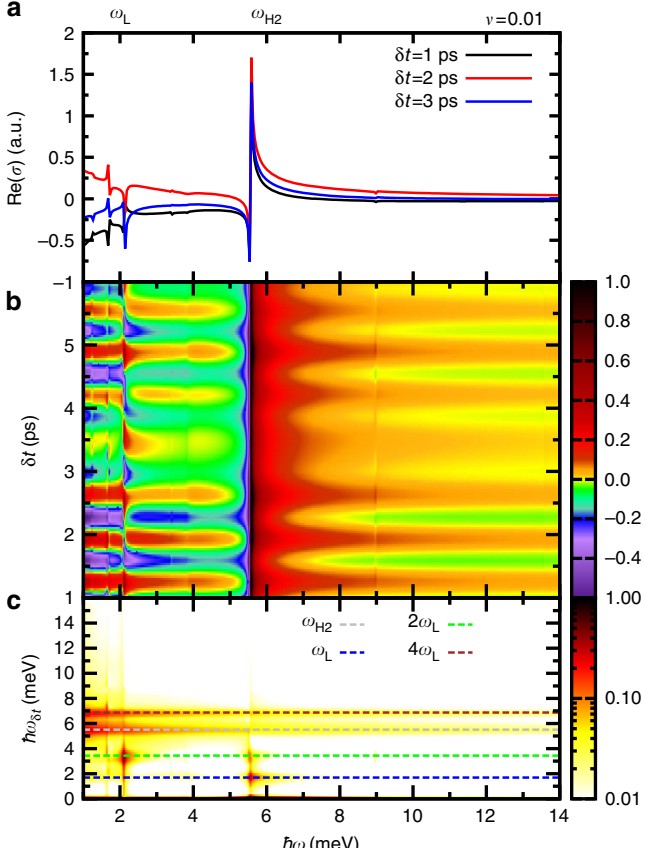

**Figure 6 | Pump–probe spectrum.** (**a**) Real part of the pump–probe response Re[$\sigma(\delta t, \omega)$] as a function of frequency $\omega$ for a two-band superconductor excited by the same pump pulse as in Fig. 2. Both the energy of the non-equilibrium Leggett mode $\omega_L$ and the Higgs mode $\omega_{H2}$ are visible in the pump–probe signal as sharp peaks. (**b**) Temporal evolution of Re[$\sigma(\delta t, \omega)$]. The intensity of the pump–probe signal is represented by the colour scale with dark red and dark violet, indicating high and low intensities, respectively. (**c**) Fourier spectrum of the pump–probe signal. The pump–probe spectrum oscillates as a function of pump–probe delay time $\delta t$ with the frequencies of the non-equilibrium Higgs mode $\omega_{H2}$ (dashed gray), the frequency of the non-equilibrum Leggett mode $\omega_L$ (dashed blue) and the frequencies of higher harmonics of the Leggett mode $2\omega_L$ (dashed green) and $4\omega_L$ (dashed red).

similar enhancement is obtained when $\omega_{H2}$ is brought into resonance with twice the frequency of the Leggett mode (Fig. 5b,d,f).

**Pump–probe signal.** Finally, let us discuss how the Higgs and Leggett modes can be observed in pump–probe spectroscopy. In view of the recent THz pump–THz probe experiments of refs 2–4, we focus on the dynamics of the optical pump–probe conductivity $\sigma(\delta t, \omega) = j(\delta t, \omega)/[i\omega A(\delta t, \omega)]$, where $\delta t$ is the delay time between pump and probe pulses, $j(\delta t, \omega)$ denotes the current density and $A(\delta t, \omega)$ represents the vector potential of the probe pulse. As the probe pulse has a weak intensity, we neglect terms of second order and higher in the probe field $A(\delta t, \omega)$. Similar to recent experiments[2–7], we take the probe pulse to be very short with width $\tau_{pr} = 0.15$ ps and centre frequency $\hbar\omega_{pr} = 5.5$ meV (see Methods). With this choice, the probe pulse has a broad spectral bandwidth such that the dynamics of the superconductor is probed over a very wide frequency range.

In Fig. 6a we plot the real part of the pump–probe response Re[$\sigma(\delta t, \omega)$] as a function of frequency for fixed $\delta t$. The non-equilibrium Leggett mode $\omega_L$ and the Higgs-mode $\omega_{H2}$ reveal themselves in the pump–probe signal as sharp peaks. Figure 6b shows Re[$\sigma(\delta t, \omega)$] versus delay time $\delta t$ and frequency $\omega$. Clear oscillations are seen as a function of delay time $\delta t$. These are most prominent at the frequencies of the lower Higgs and the Leggett modes, $\omega_{H2}$ and $\omega_L$, where $\sigma(\delta t, \omega)$ displays sharp edges as a function of $\omega$ (Fig. 6a). Fourier transforming with respect to $\delta t$ shows that the dominant oscillations are $\omega_{H2}$ and $\omega_L$ (and its higher harmonics) (Fig. 6b). We therefore predict that both the lower Higgs mode $\omega_{H2}$ and the Leggett mode $\omega_L$ can be observed in THz pump–THz probe experiments as oscillations of the pump–probe conductivity, in particular at the gap edge $2\Delta_2^\infty/\hbar$ and the Leggett mode frequency $\omega_L$. The higher Higgs mode $\omega_{H1}$, on the other hand, is not visible in the pump–probe signal, as it is strongly damped by the two-particle continuum.

## Discussion

Using a semi-numerical method based on the density matrix approach, we have studied the non-equilibrium excitation of Higgs and Leggett modes in two-band superconductors. Although the amplitude Higgs and the Leggett phase mode are decoupled in equilibrium, we find that the out-of-equilibrium excitation process leads to a strong coupling between these two collective modes. As a result, the Leggett phase mode $\omega_L$ is pushed below the Bogoliubov quasiparticle continuum and remains undamped for a wide range of interband couplings (Figs 3 and 4). Likewise, the lower Higgs mode $\omega_{H2}$ is only weakly damped, as its frequency is at the threshold to the quasiparticle continuum. To maximize the oscillatory signal of these collective modes in the pump–probe spectra, it is necessary to choose the experimental parameters as follows: (i) the pump-pulse duration $\tau$ should be smaller than the intrinsic response time of the superconductor $h/(2|\Delta_i|)$, such that the collective modes are excited in a non-adiabatic manner; (ii) the pump-pulse energy needs to be of the order of the superconducting gap (that is, in the terahertz regime), so that Bogoliubov quasiparticles are excited across the gap, but modes at higher energies $\hbar\omega \gg |\Delta_i|$ are not populated; and (iii) the pump-pulse intensity must not exceed a few nJ cm$^{-2}$, to ensure that the superconducting condensate is only partially broken up, but not completely destroyed. We have predicted that under these conditions both Higgs and Leggett modes can be observed as clear oscillations in the time-resolved pump–probe absorption spectra (Fig. 6). Similarly, we expect that collective mode oscillations are visible in other pump–probe-type experiments, for example, in time-resolved photoemission spectroscopy or time-resolved Raman scattering.

Our findings apply beyond the scope of two-band super-conductors to general collective modes in quantum materials. That is, we expect that out-of-equilibrium excitations lead to the coupling of collective modes in any symmetry-broken phase. It would be particularly intriguing to study this in more detail for the case of unconventional exotic superconductors, where several competing orders are present, such as heavy fermion superconductors or high-temperature cuprate and pnictide superconductors. In these systems the pump pulse could be used to induce a transition from one competing order to another. Furthermore, the unconventional pairing symmetries of these superconductors, such as the $d_{x^2-y^2}$-wave pairing of the cuprates, give rise to a multitude of new Higgs modes[44]. Our work indicates that pump–probe experiments will allow to coherently excite and control these novel Higgs modes, which await to be further explored both theoretically and experimentally.

## Methods

**Model definition.** The gap equations for the BCS Hamiltonian $H_{BCS}$ (see equation (1) in the main text) are given by[45]

$$\Delta_1 = \frac{1}{N} \sum_{\mathbf{k}' \in \mathcal{W}} (V_1 \langle c_{-\mathbf{k}',\downarrow,1} c_{\mathbf{k}',\uparrow,1} \rangle + V_{12} \langle c_{-\mathbf{k}',\downarrow,2} c_{\mathbf{k}',\uparrow,2} \rangle),$$

$$\Delta_2 = \frac{1}{N} \sum_{\mathbf{k}' \in \mathcal{W}} (V_2 \langle c_{-\mathbf{k}',\downarrow,2} c_{\mathbf{k}',\uparrow,2} \rangle + V_{12} \langle c_{-\mathbf{k}',\downarrow,1} c_{\mathbf{k}',\uparrow,1} \rangle),$$

(3)

where $N$ is the number of lattice points, $V_1$ and $V_2$ denote the intraband interactions and $V_{12} = v V_1$ is the interband coupling. The two-band superconductor is brought out of equilibrium via the coupling to a pump pulse, which is modelled by

$$H_{Laser} = \frac{e\hbar}{2} \sum_{\mathbf{k},\mathbf{q},\sigma,l} \frac{(2\mathbf{k}+\mathbf{q})\mathbf{A}_{\mathbf{q}}(t)}{m_l} c^{\dagger}_{\mathbf{k}+\mathbf{q},\sigma,l} c_{\mathbf{k},\sigma,l}$$

(4)

$$+ \frac{e^2}{2} \sum_{\mathbf{k},\mathbf{q},\sigma,l} \frac{(\sum_{\mathbf{q}'} \mathbf{A}_{\mathbf{q}-\mathbf{q}'}(t)\mathbf{A}_{\mathbf{q}'}(t))}{m_l} c^{\dagger}_{\mathbf{k}+\mathbf{q},\sigma,l} c_{\mathbf{k},\sigma,l},$$

where $m_l$ is the effective electron mass of the $l$th band and $\mathbf{A}_{\mathbf{q}}(t)$ represents the transverse vector potential of the pump laser. We consider a Gaussian pump pulse described by

$$\mathbf{A}_{\mathbf{q}}(t) = \mathbf{A}_0 e^{-(\frac{2\sqrt{\ln 2}t}{\tau})^2} (\delta_{\mathbf{q},\mathbf{q}_0} e^{-i\omega_0 t} + \delta_{\mathbf{q},-\mathbf{q}_0} e^{i\omega_0 t}),$$

(5)

with central frequency $\omega_0$, pulse width $\tau$, light-field amplitude $\mathbf{A}_0 = |\mathbf{A}_0|\hat{\mathbf{e}}_y$ and photon wave vector $\mathbf{q}_0 = q_0 \hat{\mathbf{e}}_x$.

**Density matrix formalism.** To simulate the non-equilibrium dynamics of the two-band superconductor (1), we use a semi-numerical method based on the density matrix formalism. This approach involves the analytical derivation of equations of motions for the Bogoliubov quasiparticle densities $\langle \alpha^{\dagger}_{\mathbf{k},l} \alpha_{\mathbf{k}',l} \rangle$, $\langle \beta^{\dagger}_{\mathbf{k},l} \beta_{\mathbf{k}',l} \rangle$, $\langle \alpha^{\dagger}_{\mathbf{k},l} \beta^{\dagger}_{\mathbf{k}',l} \rangle$ and $\langle \alpha_{\mathbf{k},l} \beta_{\mathbf{k}',l} \rangle$, which are then integrated up numerically using a Runge Kutta algorithm. The Bogoliubov quasiparticle densities are defined in terms of the fermionic operators $\alpha_{\mathbf{k},l}$ and $\beta_{\mathbf{k},l}$, with

$$\alpha_{\mathbf{k},l} = u_{\mathbf{k},l} c_{\mathbf{k},l,\uparrow} - v_{\mathbf{k},l} c^{\dagger}_{-\mathbf{k},l,\downarrow},$$

(6)

$$\beta_{\mathbf{k},l} = v_{\mathbf{k},l} c^{\dagger}_{\mathbf{k},l,\uparrow} + u_{\mathbf{k},l} c_{-\mathbf{k},l,\downarrow},$$

(7)

where $v_{\mathbf{k},l} = \Delta_l(t_i)/|\Delta_l(t_i)| \sqrt{(1-\epsilon_{\mathbf{k},l}/E_{\mathbf{k},l})/2}$, $u_{\mathbf{k},l} = \sqrt{(1+\epsilon_{\mathbf{k},l}/E_{\mathbf{k},l})/2}$ and $E_{\mathbf{k},l} = \sqrt{\epsilon^2_{\mathbf{k},l} + |\Delta_l(t_i)|^2}$. We emphasize that the coefficients $u_{\mathbf{k},l}$ and $v_{\mathbf{k},l}$ do not depend on time, that is, the temporal evolution of the quasiparticle densities is computed with respect to a fixed time-independent Bogoliubov-de Gennes basis in which the initial state is diagonal. The equations of motion for the quasiparticle densities are readily obtained from Heisenberg's equation of motion. Since equation (1) represents a mean-field Hamiltonian, this yields a closed system of differential equations and hence no truncation is needed (for details, see refs 17,29,23,25).

**Pump–probe response.** All physical observables, such as the current density $\mathbf{j}_{\mathbf{q}_{pr}}(\delta t, t)$, can be expressed in terms of the quasiparticle densities. For the current density we find that

$$\mathbf{j}_{\mathbf{q}_{pr}}(\delta t, t) = -e\hbar \sum_{\mathbf{k},l,\sigma} \frac{2\mathbf{k}+\mathbf{q}_{pr}}{2m_l V} \langle c^{\dagger}_{\mathbf{k},l,\sigma} c_{\mathbf{k}+\mathbf{q}_{pr},l,\sigma} \rangle (\delta t, t)$$

$$- e^2 \sum_{\mathbf{k},l,\mathbf{q},\sigma} \frac{\mathbf{A}_{\mathbf{q}_{pr}-\mathbf{q}}}{m_l V} \langle c^{\dagger}_{\mathbf{k},l,\sigma} c_{\mathbf{k}+\mathbf{q},l,\sigma} \rangle (\delta t, t),$$

where $\mathbf{A}_{\mathbf{q}_{pr}}(\delta t, t)$ and $\mathbf{q}_{pr} = |\mathbf{q}_{pr}|\hat{\mathbf{e}}_x$ are the vector potential and the wave vector of the probe pulse, respectively. With this, we obtain the pump–probe conductivity via[23,46]

$$\sigma(\delta t, \omega) = \frac{j(\delta t, \omega)}{i\omega A(\delta t, \omega)},$$

(9)

where $j(\delta t, \omega)$ and $A(\delta t, \omega)$ denote the Fourier transformed $y$ components of the current density $\mathbf{j}_{\mathbf{q}_{pr}}(\delta t, t)$ and the vector potential $\mathbf{A}_{\mathbf{q}_{pr}}(\delta t, t)$, respectively. To compute the effects of the probe pulse, we neglect terms of second order and higher in the probe field $A_{pr}(t)$, as the probe pulse has a very weak intensity.

**Numerical discretization and integration.** To keep the number of equations of motion manageable, we have to restrict the number of considered points in momentum space. The first restriction is that we only take expectation values with indices $\mathbf{k}$ and $\mathbf{k}+\mathbf{q} \in \mathcal{W}$ into account. This means that we concentrate on the $\mathbf{k}$-values where the attractive pairing interaction takes place. Furthermore, as the external electromagnetic field may add or subtract only momentum $n\mathbf{q}_0$, it is sufficient to consider expectation values with indices $(\mathbf{k}, \mathbf{k}+n\mathbf{q}_0)$, where $n \in \mathbb{Z}$. For small amplitudes $|\mathbf{A}_{\mathbf{q}_0}|$ the off-diagonal elements of the quasiparticle densities decrease rapidly as $n$ increases, as $(\mathbf{k}, \mathbf{k}+n\mathbf{q}_0) = O(|\mathbf{A}_{\mathbf{q}_0}|^{|n|})$. Thus, we set all entries with $n > 4$ to 0. With this momentum–space discretization, we obtain the order of $10^5$ equations, which we are able to solve numerically using high-efficiency parallelization. Further, technical details can be found in refs 23,25.

**Data availability.** All relevant numerical data are available from the authors upon request.

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

## Acknowledgements

We gratefully acknowledge many useful discussions with A. Avella, S. Kaiser and R. Shimano. G.S.U. and H.K. acknowledge financial support by the DFG in TRR 160 and by the Stiftung Mercator. H.K. thanks the Max-Planck-Institut FKF Stuttgart for its hospitality.

## Author contributions

The density matrix simulations were developed and run by H.K., N.B. and A.P.S. All authors contributed to the discussion and interpretation of the results, and to the writing of the paper.

## Additional information

**How to cite this article**: Krull, H. *et al.* Coupling of Higgs and Leggett modes in non-equilibrium superconductors. *Nat. Commun.* 7:11921 doi: 10.1038/ncomms11921 (2016).

