## [Peer Review File · Nature Communications]

REVIEWERS' COMMENTS:

Reviewer #1 (Remarks to the Author):

A. This paper treats nonequilibrium excitations in two-gap superconductors which can be elicited by optical pump-probe experiments. The authors show that collective amplitude (Higgs) and phase (Leggett) modes are excited and coupled out of equilibrium, in contrast to their decoupled behavior in equilibrium systems.

B. This work is very timely. As the authors stress, most theoretical work to date has focused on nonequilibrium collective modes in single-gap superconductors. The current work should have relevance to materials such as MgB₂ and the iron pnictide class.

C. The model is somewhat standard and is well-explained. From what I can tell from the authors' description, the self-consistent BCS gap equations are solved at each temporal step and the pump-probe process is simulated using a density matrix formalism. The details of the model and the authors' method of solution are very clearly explained in the Methods section. In general, this paper is an admirable model of clarity.

D. No error bars are provided on the numerical results, surely because they can be achieved to very high precision. The discussion and display of the numerical data is highly professional.

E. The conclusions are physically sound and in my view likely robust. Indeed the coupling between Higgs and Leggett modes under nonequilibrium conditions is to be expected. Still, the novelty of this investigation, the clarity of the presentation, and the relevance to experiment make this a compelling paper.

F. It might be pedagogically interesting to track the behavior of this model as one of the gaps is taken to infinity.

G. The references are extensive and appropriate.

H. As stated above, this paper is wonderfully clear and should be of substantial interest to a fairly broad range of condensed matter physicists working in nonequilibrium superconductivity.

Reviewer #2 (Remarks to the Author):

The article of H. Krull et al. deals with the collective modes induced by a spontaneous broken symmetry in condensed matter.

Here H. Krull et al. develop a semi numerical calculation based on the density matrix formalism and show that both Higgs and Leggett collective modes can be detected by optical pump probe experiments in a two-gap superconductor out of equilibrium.

This is a consequence of a non adiabatic excitation process which leads to a direct coupling between the phase and amplitude modes which are decoupled in equilibrium system. Moreover the authors

propose a new route for probing and coherently control the Higgs and Leggett modes.

This theoretical article is very well written and understandable to non-specialists on the subject.

The authors have a courageous step in the sense that it offers a predictive theoretical works and offers an unique experience to check their predictions.

The results are novel. From my point of view, this article deserves to be publishing in a journal such as Nature Communications.

The authors present very exciting physics.

Therefore I strongly support this article.

I propose some minors clarifications:

page 1 ,2nd column:

In the caption of figure 1 (a) , precise the meaning of the red and orange curves (oscillation of the order parameter and its monotonically decrease)

page 3, 1st column:

I would like the authors precise if the choices of $\Delta_1=7$ meV and $\Delta_2=3$ meV correspond to some specific two-gap superconductor? MgB₂?

Reviewer #3 (Remarks to the Author):

As someone involved in experiments that were shown by others to be the Higgs mode in a single-gap superconductor and who carried out theoretical work on Leggett modes of a two-gap superconductor, I find this theoretical/computational work quite interesting. The authors show that use of optical pump-probe experiments can excite and indeed control both kinds of mode, putting them temporarily out of equilibrium and controlling the superconductor order parameter.

The figures are essential to showing the results of the simulations, but I had to "blow up" Figure 1 by 50% or more to really understand it.

My recommendation is to publish this work.

I. REPLY TO REFEREE REPORTS

We would like to thank all three referees for their positive reports. Here is our response to their comments and recommendations.

A. Reviewer #1

“A. This paper treats nonequilibrium excitations in two-gap superconductors which can be elicited by optical pump-probe experiments. The authors show that collective amplitude (Higgs) and phase (Leggett) modes are excited and coupled out of equilibrium, in contrast to their decoupled behavior in equilibrium systems.

B. This work is very timely. As the authors stress, most theoretical work to date has focused on nonequilibrium collective modes in single-gap superconductors. The current work should have relevance to materials such as MgB₂ and the iron pnictide class.

C. The model is somewhat standard and is well-explained. From what I can tell from the authors’ description, the self-consistent BCS gap equations are solved at each temporal step and the pump-probe process is simulated using a density matrix formalism. The details of the model and the authors’ method of solution are very clearly explained in the Methods section. In general, this paper is an admirable model of clarity.

D. No error bars are provided on the numerical results, surely because they can be achieved to very high precision. The discussion and display of the numerical data is highly professional.

E. The conclusions are physically sound and in my view likely robust. Indeed the coupling between Higgs and Leggett modes under nonequilibrium conditions is to be expected. Still, the novelty of this investigation, the clarity of the presentation, and the relevance to experiment make this a compelling paper.”

We thank the referee for his/her positive report.

F. It might be pedagogically interesting to track the behavior of this model as one of the gaps is taken to infinity.

This is a very interesting suggestion. When one of the gaps is taken to infinity, the pump

pulse will only excite quasiparticles across the other gap, whose value is finite. Hence, the system will behave like a non-equilibrium one-gap superconductor and the Leggett mode will not be excited. We leave the detailed study of this question for future research.

G. The references are extensive and appropriate.

H. As stated above, this paper is wonderfully clear and should be of substantial interest to a fairly broad range of condensed matter physicists working in nonequilibrium superconductivity.

We thank the referee for these very positive comments on our work.

B. Reviewer #2

This theoretical article is very well written and understandable to non-specialists on the subject. The authors have a courageous step in the sense that it offers a predictive theoretical works and offers an unique experience to check their predictions. The results are novel. From my point of view, this article deserves to be publishing in a journal such as Nature Communications. The authors present very exciting physics. Therefore I strongly support this article.

We thank the referee for these very positive comments.

I propose some minors clarifications: page 1, 2nd column: In the caption of figure 1 (a) , precise the meaning of the red and orange curves (oscillation of the order parameter and its monotonically decrease)

We thank the referee for this suggestion. We have added a sentence in the caption of Fig. 1 to clarify the meaning of the red and orange curves.

page 3, 1st column: I would like the authors to precise if the choices of $\Delta_1 = 7$ meV and $\Delta_2 = 3$ meV correspond to some specific two-gap superconductor? MgB2?

Yes, indeed, the choice for these parameters is motivated by MgB_2 , where the gap for the σ band is about 7 meV, while on the π band the gap is about 3 meV. We have added a sentence on page 3 to clarify the reason for choosing these gap values.

C. Reviewer #3

As someone involved in experiments that were shown by others to be the Higgs mode in a single-gap superconductor and who carried out theoretical work on Leggett modes of a two-gap superconductor, I find this theoretical/computational work quite interesting. The authors show that use of optical pump-probe experiments can excite and indeed control both kinds of mode, putting them temporarily out of equilibrium and controlling the superconductor order parameter.

We thank the referee for these positive remarks on our work.

The figures are essential to showing the results of the simulations, but I had to "blow up" Figure 1 by 50% or more to really understand it.

As suggested by the referee, we have increased the size of Fig. 1, and also increased the font size of the labels.

D. Summary of changes made

1. Page 1, abstract:

We have added the sentence "The direct detection of ...".

2. Page 2, introduction:

We have added the sentence "In particular, this technique has been used to ...". We have moved the discussion of Fig. 1 to the Results section.

3. Page 3, results:

We have added the paragraph "In pump-probe measurements the ...".

4. Page 4, results:

We have added the sentence “Motivated by the numbers for MgB2 [43], ...”.

5. Page 4, references:

We have modified the references, such that they now comply with the nature style guide. We have added reference [43].

6. Page 13, Data availability:

We have added the sentence “All relevant numerical data are available ...”.

7. Figures:

We have modified the labels in Figs. 1, 2, 3, 5, and 6.